# Benchmarking of Various LiDAR Sensors for Use in Self-Driving Vehicles in Real-World Environments

**DOI:** 10.3390/s22197146

**Published:** 2022-09-21

**Authors:** Joschua Schulte-Tigges, Marco Förster, Gjorgji Nikolovski, Michael Reke, Alexander Ferrein, Daniel Kaszner, Dominik Matheis, Thomas Walter

**Affiliations:** 1Mobile Autonomous Systems and Cognitive Robotics Institute, FH Aachen—Aachen University of Applied Sciences, 52066 Aachen, Germany; 2Hyundai Motor Europe Technical Center GmbH, 65428 Rüsselsheim am Main, Germany

**Keywords:** LiDAR, benchmark, self-driving

## Abstract

In this paper, we report on our benchmark results of the LiDAR sensors Livox Horizon, Robosense M1, Blickfeld Cube, Blickfeld Cube Range, Velodyne Velarray H800, and Innoviz Pro. The idea was to test the sensors in different typical scenarios that were defined with real-world use cases in mind, in order to find a sensor that meet the requirements of self-driving vehicles. For this, we defined static and dynamic benchmark scenarios. In the static scenarios, both LiDAR and the detection target do not move during the measurement. In dynamic scenarios, the LiDAR sensor was mounted on the vehicle which was driving toward the detection target. We tested all mentioned LiDAR sensors in both scenarios, show the results regarding the detection accuracy of the targets, and discuss their usefulness for deployment in self-driving cars.

## 1. Introduction

While first car manufacturers are receiving approvals for SAE level 3 self-driving functions (see [1,2]), the whole industry is making huge progress to enter the era of automated driving. A self-driving car needs to perceive its environment through its sensors, interpret the sensed data, plan and decide future actions to take and finally perform the chosen actions. Despite some prominent opinions that the only required sensor type are vision-based sensors, many manufacturers see the need for the use of further sensor technologies in order to increase the accuracy and reliability of the vehicle-surrounding perception required for higher levels of automation. They investigate other sensors, such as RADAR or LiDAR [3], for perceiving the environment of the self-driving car [4]. Additionally, LiDAR sensors have proven their usefulness in academic self-driving projects, such as the DARPA Grand Challenge [5], which raises vehicle OEMs interest for use in series development as well.

In this paper, a benchmark of currently available non-rotary LiDAR sensors is presented to identify suitable LiDAR hardware for real-world usage. The selected LiDARs were tested in our automated driving (AD) platform, which was developed in a long-standing research cooperation between the Hyundai Motor Europe Technical Center GmbH (HMETC) and the University of Applied Science (FH) Aachen. The overall goal is to develop and to integrate our automated driving software framework [6] into a HMETC prototype vehicle to support the European research activities of HMETC, particularly in the European project Hi-Drive [7].

One common argument against LiDAR sensors for mass-produced cars is their price, which usually heavily exceeds the price of vision-based systems. Current technical developments, such as the development of solid-state LiDARs, have further reduced the price of these systems, making them more attractive for vehicle OEM series development activities. The systems presented in this paper, for instance, start at around USD 1300. We present and evaluate four different test scenarios, which can be translated into real-world use cases:

Scenario 1: Detecting sphere targets in increasing distances;Scenario 2: Detecting a square meter reference plane;Scenario 3: Detecting sphere targets in motion;Scenario 4: Detecting other vehicles in motion.

Based on the use case, the test scenario results can be weighted differently: For instance, when performing a slow manoeuver in a parking space, a sensor which is good at a close distance with a high accuracy is required, while for highway driving, it is more important to have a high range; as for the accuracy of the sensor in this scenario, accuracies exceeding 1 cm is acceptable, while higher accuracies might be required on a car park. A more detailed description of the different scenarios is given in Section 3.

Our results show that the different sensors perform quite differently in the various test scenarios. Some perform well for static scenarios, while they fail at the dynamic scenarios; some have high deviations in measuring depth information while others struggle with precise measurements for *x* and *y* ranges. We further investigated the observed phenomena, and our findings show that this also depends on the scanning patterns that the different sensors use. Other observations showed that some sensors have particular problems on object edges, such as crash or jersey barriers. The main contributions of this paper are as follows:To propose a LiDAR benchmark in realistic drive scenarios.To find particular relationships between the scan patterns and the performance in the proposed real-word tests.To come up with a list of state-of-the-art scanning devices for self-driving cars.

The paper is organized as follows: In Section 2, we discuss related research, while in Section 3, we define four different benchmarks, present the results and discuss them in Section 4 and Section 5, respectively. We conclude with Section 6.

## 2. Related Work

Recently, LiDAR sensors have gained more attention from the scientific community, as many research results about perception [8] and localisation [9,10] algorithms have been published.

The team of Lambert, Carballo et al. [11] followed a similar approach and shared in their work similar ideas. They analyzed 10 different LiDARs. Similar to our work, they designed their tests with multiple detection targets and varying distance as well as the surface reflectivity of the objects being measured. Furthermore, they used a self-calibration method for each LiDAR. However, their research focuses solely on static scenarios, whereas we also study how the tested LiDAR sensors perform when the vehicle is in motion passing a still-standing target better reflecting real-world situations. By doing so, one of our findings shows that the sensors should be calibrated differently when deploying them in static or dynamic use cases. We discuss this issue in depth in Section 5. This benchmark is focused on solid-state LiDARs instead of rotary 360° LiDARs.

In contrast to our approach with designing tests suited for real-world scenarios, Cattini, Cassaneli et al. [12] proposed a different test setup. They introduced a procedure to create a very repeatable and precise testing setup for LiDAR sensors. It constraints multiple motions of displacement and rotation of the sensor and the measurement targets to a single dimension, which allows for a more precise setup of the tests. It also incorporates camera data for the evaluation. While the precision and repeatability is important, the scalability and variability of such constrained testing setups is, however, very limited.

Bijelic, Gruber, Ritter et al. benchmarked LiDARs in a fog chamber [13]. Their goal was to evaluate the performance of LiDARs when being exposed to varying amounts of fog. They tested four sensors from two major manufacturers. Their evaluation was based on comparing the birds-eye-view representation of point clouds and the intensities of parts of point clouds recorded from a fixed position at different exposure levels of fog.

The International Society of Optics and Photonics has also drafted a set of tests and specifications on how to execute and evaluate the tests [14]. The draft recommends testing LiDARs on functions with high security implications, such as the detection of children-sized targets or influence on human eyesight when exposed to LiDARs.

As shown in [8], in particular, the field of deep learning for 3D object detection based on LiDAR data has made major leaps forward by yielding higher accuracies than previous methods. Some of the best performing algorithms in terms of quality and execution times are algorithms processing only point cloud information, which often are provided by a LiDAR. Good examples for this are PV-RCNN [15], PointPillars [16] and PointRCNN [17], to name just a few. Despite the mentioned approaches use different methods, they have in common that they all extract features from the point cloud: (1) PC-RCNN uses a combination of voxel- and point-based feature extractions; (2) PointPillar deploys a feature extraction from pillar-like groupings of points in a pseudo-image representation of a point cloud; and (3) relies solely on a point-based feature extraction. The three networks show significantly different performances when faced with different properties in the underlying point cloud data [18]. Properties influencing the performance outcome include noisiness and density. Both are properties defined by a selected sensor for testing and environmental influences.

For the field of perception-based localization, Qin Zou, Qin Sun, Long Chen et al. [10] showed a good comparison of many of the bleeding-edge algorithms in said field. The algorithms utilize such methods as Kalman filters, particle filters and loop closure to achieve the goal of localizing an agent traversing unknown terrain, while also creating maps of the surroundings.

For the verification and presentation of the performance, researchers usually test their algorithms on publicly available LiDAR-based point-cloud data sets. Benchmarking data sets, such as Kitti [19], NuScenes [20] and Waymo [21], are currently very popular in this regard. Each provides a large number of (labeled/classified) point cloud data for research and development. Although the quantity of records in those data sets is large, and the variety in the recorded surroundings and objects is sufficient, each of the benchmark data sets only uses one specific LiDAR sensor from one manufacturer to record the data. The LiDAR sensors used in all of the mentioned data sets is of the same type—A 360° rotary LiDAR. In general, it is a good idea to make use of the same type of LiDAR sensors for the different data sets, as the results of the benchmarks can be compared more easily, as they are not prone to calibration errors or different calibrations of the sensors. This approach is beneficial for the testing and evaluation of 3D point cloud-based algorithms. To investigate the practical use of different LiDAR sensors, this approach needs to be reversed: Different LiDAR sensors need to be evaluated in the exact same use case with the same algorithms. The algorithms we settled on to evaluate our use cases are RANSAC [22] for estimation of the position of geometric shapes in point clouds and a ray-based ground–object segmentation derived from the ideas from Petrovskaya and Thrun [23].

## 3. Benchmark Concept

This section will give an overview of the benchmark concepts. In Section 3.1, we present the requirements which are derived from real-world scenarios and show our test candidates. In Section 3.2, Section 3.3, Section 3.4 and Section 3.5, we introduce the different LiDAR targets (spheres and planes) that were used during the tests and had to be detected by the sensors. Finally, details about the deployed detection algorithms will be discussed.

### 3.1. Test Candidates

HMETC selected several state-of-the-art LiDAR sensors which were available on the market or at a close-to-market series production stage. The selected sensors are either mechanical (moving/rotating mirror or lens parts) or solid-state types, based on micro-mirrors with MEMS (Micro-Electro-Mechanical Systems) technology. Flash LiDARs and optical phased array LiDARs could not be tested at the time of writing of the paper because some of the sensors either exceeded the price limit requirement or were ruled out due to other HMETC requirements, such as avoiding suppliers affected by chip shortages and sensors still in development or in the validation phase.

In our study, we mainly focused on non-rotary LiDARs for the reason that those systems do not have moving parts, which are prone to wear and tear or are more sensitive to challenging environmental conditions. It is planned to integrate the LiDARs into the vehicle front grill to have a certain height above the ground, compared to a mounting in the bumper below the grill. This allows to scan a wider field of view in front of the vehicle to support object perception and localization use cases during vehicle movement (speeds from 30 km/h–80 km/h). This is a compromise in the design philosophy to balance the perception needs and the impact that the mounting of the sensor in the vehicle has. It is also related to the fact that HMETC test vehicle is a step-by-step implementation to achieve the level of fully autonomous driving. These considerations make the mostly roof-mounted 360° mechanically spinning LiDARs known, for instance, from Waymo’s [21] robotaxis, unsuitable, oversized and overpriced. Based on the described usage conditions and the hardware requirements given below, the sensors shown in Table 1 are tested.

We first start with the different requirements for the LiDAR sensors:

*Field of View.* We aim at a horizontal field of view of more than 100° and a vertical field of view of more than 10° to reduce the number of sensors that has to mounted on the vehicle.*Minimal detection distance and range.* The distance to detect the LiDAR targets should be less than 3 m while the range of the LiDAR should be more than 120 m.*Resolution and number of scan lines.* The sensors should of course have a high resolution below 0.4° and at least five scan lines to be able to detect the LiDAR targets and real-world objects.*Update rate or frame rate.* In order to avoid longer delays in the object detection, the sensor systems should have an update frequency or frame rate of more than 5 Hz.*ROS/ROS2 support.* For an easy integration into our control software stack [6], a Linux-based system implementation and an AD framework based on ROS2 is preferred.*Robustness of sensor systems.* The test candidates should work well also in tougher weather conditions, and the sensor performance should not notably degrade under those conditions.

The LiDARs were evaluated using multiple distinctive test cases, reflecting the later real-world usage scenarios (outdoors, urban environment, dynamic movement, and on-road). This minimizes the selection of a false-positive-rated sensor which performs well in a laboratory environment bench test, but shows issues with, for instance, road surface reflections, ghost objects such as plants or leaves, or blinding sunlight when being used in real-world environments. Mounting each sensor to a test vehicle can also reveal possible negative impacts, such as (electromagnetic) interference with other test vehicle components, including measurement units or ECU-related signal transmission delays, such that the performance of a sensor might degrade to a level not acceptable in an automotive environment. The sensors in Table 1 were selected based on the following criteria:Packaging size, interface and cabling;Sensor IP rating: Robustness against water, rain, snow, mud, stonesSetup time, run-time without failure;Power consumption: Low power consumption (less than 15 W per unit);Sensor availability;Manufacturer support;Configuration options;Scan pattern to ensure similar test conditions.

The sensors were provided by HMETC for running the benchmark activities.

Most sensors have the possibility to change FPS (frames per second) and/or points per frame. Some even allow changes in the scanning pattern. For our tests, we selected a number of LiDARs with very specific patterns that moreover meet the requirements presented in the previous section. In the next sections, we introduce how we detect the different LiDAR targets.

### 3.2. Scenario 1: Static Spheres in Increasing Distances

In this section, the first test scenario is presented, where a triangular-shaped sphere target has to be detected by the LiDAR system in different distances, ranging from 7 m–25 m. The vehicle is not in motion for this scenario.

The deployed LiDAR target is shown in Figure 1. A structure of three spheres aligned in an equilateral triangle is used. Each sphere has a diameter of 40 cm. The space between each sphere center point is approximately 110 cm. To detect the LiDAR target, RANSAC [22] is being applied to find the spheres inside the point cloud. Each time the algorithm detects exactly all three spheres, the centroid of the triangle is calculated, containing the measured distances in the point cloud. The process repeats *n* times; the result is a dataset with *n* values for the centroid (see Figure 1a) because a measurement was only added to the dataset if all the spheres were detected.

In this scenario, we measure the performance of the LiDAR sensors for increasing distances, while the distance does not change during the measurement, i.e., LiDAR sensor and measured objects are standing still. Four different distances between the LiDAR sensor and the sphere construction were used: 7 m, 12 m, 15 m, and 25 m. The minimum distance was chosen because at 7 m all LiDAR had all spheres in their FOV. The maximal distance was chosen after noticing that at this range the first sensors were unable to detect the spheres; knowing the performance at 25 m was a good indication for the following dynamic scenarios. The decision to use 12 m and 15 m in between instead of equidistant distances has no particular reasons. For each distance and for each LiDAR sensor, 1000 data points were taken (cf. Section 4.1 for the results).

### 3.3. Scenario 2: Static Square Meter Reference Plane

In the second scenario, errors that might have been introduced by deploying RANSAC for the sphere detection in Scenario 1 should be avoided and should justify the validity of the other scenarios. If a deployed detection algorithm favours or disfavours a particular LiDAR sensor, can be identified with this scenario. We constructed a free-floating square meter plane as shown in Figure 2a. The wooden square meter plane was set up 7 m in front of the LiDAR sensors (both standing). As the laser beams will just hit the front of the square meter plane, the plane will create free floating points inside the point cloud. Around these points, a cut-out box is placed (virtually inside the point cloud), and all points that are not inside this box are extracted from the point cloud (Figure 2b). The remaining points inside the point cloud are just the points originated by the reflection of the square meter plane itself. Inside the remaining point cloud, the minima and maxima in every direction can be determined, and we can be establish the following equations:$$\begin{array}{c} {\mathrm{zAxis}}_{max}-{\mathrm{zAxis}}_{min}\approx 1m\\ {\mathrm{yAxis}}_{max}-{\mathrm{yAxis}}_{min}\approx 1m\\ {\mathrm{xAxis}}_{max}-{\mathrm{xAxis}}_{min}\approx 0m \end{array}$$

So, we expect to measure the width and the height of the plane ( 1 m in *z* and *y* dimension). Of course, the difference in the distance to the plane should be 0 (*x* dimension). Refer to Figure 2c,d for the naming of the axes. The measured sizes will scatter around the true size and with 1000 measurements we approximate a normal distribution of the values just like with Scenario 1. The results are shown in Section 4.2.

### 3.4. Scenario 3: Dynamic Spheres

This scenario is used to measure the performance of the LiDAR sensors for decreasing distances, while the LiDAR moves toward the statically placed sphere construction during the measurement (dynamic measurement), as shown in Figure 3. The LiDAR sensor was mounted on top of a test vehicle The vehicle started 300 m away from the measurement target, and drove automatedly toward the sphere construction with a speed of 10 km/h. The speed of 10 km/h was chosen for both dynamic scenarios (Scenario 3 and 4) to cater for the update rates of the LiDAR sensor. Other tests with 80 km/h were made, but were later excluded from both dynamic scenarios because of the distortion problem which will be discussed in Section 4.3. For each measurement, the vehicle drove at a constant speed. The position of the sphere construction and the position of the test vehicle was measured by GPS. Our detection algorithm measured the distance to the triangular sphere construction ($distance_{lidar}$) at driving distance *x*. As a reference, the distance between the vehicle’s GPS position and the triangular sphere construction’s GPS position was calculated as $distance_{gps}$. Now, the difference between the distance measured by LiDAR and by GPS at any given distance can be calculated as:(1)$$distance_{\Delta }\left(x\right)=distance_{lidar}\left(x\right)-distance_{gps}\left(x\right)$$

Note that $distance_{\Delta }$ will not become zero due to the offset between the vehicle’s GPS position the position of the LiDAR sensor.

### 3.5. Scenario 4: Dynamic Vehicle Detection

This scenario is designed to be close to a real-world use case. This scenario is designed analogously to Scenario 3, but instead of the sphere construction, another still-standing vehicle was placed on the shoulder lane and was to be detected by the LiDAR sensor (see Figure 4). Again as in Scenario 3, we used Equation (Equation 1) to measure the offset between the true position and the detected position of the target.

The algorithms used in the previous scenarios could not be used for object detection, because it is only capable of detecting spheres. Therefore, another algorithm had to be developed. Using a ray-based ground object segmentation [23], the algorithm converts the point cloud into a 2D grid of cells. For each cell of the grid, the algorithm calculates a height profile (see Figure 5). The height of a cell is equal to the highest measured point of the point cloud inside the grid. Once each cell has a height profile, the cell extension starts. Starting from the origin and then going up-stream (in the driving direction, where the bottom of the grid is nearest to the vehicle), the algorithm calculates an angle between two grid cells (see side view of Figure 5). The calculated angle is compared to an angle threshold. If the angle between two cells is bigger than the threshold, the cell is flagged as non-ground. This means it contains a whole object or only a part of an object bigger than the cell. In the end, the resulting point cloud with all the non-ground-flagged cells is split into single objects by extracting the resulted cluster. The results of Scenario 4 are shown in Section 4.4. Additionally, for the difference between the measured distance by LiDAR and the measured distance by GPS, also the width and the height of the second vehicle were measured with the LiDAR.

## 4. Results

This section will show the results and will give insights on the comparison, the decisions that were made, and some compelling problems that occurred. In all scenarios, multiple datasets were captured. A dataset for one measurement will scatter around an average, resulting in a normal distribution. The static scenarios are evaluated by comparing standard deviations. For dynamic scenarios, the measured points correlate with the distance.

### 4.1. Scenario 1: Static Spheres Increasing Distances

Figure 6 shows the standard deviations according to each axis and distance. The Blickfeld Cube as well as the Innoviz Pro failed to perform in this scenario at a distance of 25 m; hence, the decision was made to exclude both of them from further investigations in dynamic scenarios. Even though the Blickfeld Cube Range yielded good results, it was excluded for dynamic scenarios as well, due to its too narrow field of view.

Despite being accurate at the 7 m distance, the *Blickfeld Cube* sensor shows less accuracy at higher distances and fails to measure at 25 m. This is due to its small amount of available scan points per frame. It is noticeably lower than all the others (see data sheets in Table 1). The *Blickfeld Cube Range* on the other hand shows the highest accuracy in this test due to its zoom lens. At a distance of 15 m it is about as accurate as the Blickfeld Cube at 7 m. The zoom lens helps to keep the beams close together resulting in a dense scanning pattern even in higher distances. However, the dense scanning pattern does, on the other hand, result in a too narrow FOV and therefore the sensor was excluded from dynamic tests. The *Velodyne Velarray* performed with an average result. There is a noticeable decrease in the accuracy between 7 m to 12 m. Compared to its competitors, the accuracy of the *Robosense M1* was decreasing less with higher distances. The reason for this could be its really high amount of points and homogeneous scan pattern with the same amount of beams in vertical and horizontal alignment. Finally, the *Livox Horizon* provided a high accuracy at 7 m to 15 m distance, while at a distance of 25 m, the accuracy decreases. Fewer points hitting the sphere at this distance lead to a smaller accuracy. It shows marginal noise for each point; this also explains the accurate measurements at closer distances.

### 4.2. Scenario 2: Static Square Meter Reference Plane

Figure 7 shows the measured standard deviations for each LiDAR in *x*, *y* and *z* direction for this scenario. The *x*-axis refers to the distance to the square meter plane, the *y*-axis to the width and the *z*-axis to the height of the plane. With the Blickfeld Cube and the Velarray sensors, we observed a low accuracy for the distance measurement (*x*-axis). We explain this observation in detail in the next paragraph and will refer to it as the *edge fringing problem* in the rest of the paper. The problem did not occur with the Velarray or the Blickfeld Cube.

In Figure 8, we explain the edge fringing problem. As can be noticed on the edges of the plate, points are falsely detected behind the plate. This effect was strongest at the top and the bottom (see Figure 8). This phenomenon was observed with almost all sensors. In order to figure out what was causing that problem, a sensor was turned 90 degrees on its *x*-axis while scanning the plate. Prior to the rotation, the problem was seen at the top and the bottom; after the rotation, the problem was seen on the sides of the plate. This means it was caused by the sensor’s scan pattern, not the wooden plate itself. This test was repeated with another sensor showing the same behavior.

The *Blickfeld Cube* showed a higher accuracy in *x*-axis measurements than its competitors, while showing bad results for the *y* and *z*-axis accuracy. Despite having the same hardware, the *Blickfeld Cube Range* did not yield the same results as the Blickfeld Cube. The accuracy of *y*- and *z*-axis are noticeably more accurate than the *x*-axis measurements, which can be traced back to the edge fringing problem. The *Innoviz Pro* showed less accurate *x*-axis measurements, again because of the edge fringing problem, while the *Velodyne Velarray* achieved the best results, providing the most accurate measurements for all three axis. The sensor suffering the most from the edge fringing problem was the *Robosense M1*, while the *Livox Horizon* performed with an average accuracy despite the edge fringing problem.

### 4.3. Scenario 3: Dynamic Spheres

During the measurements, a problem was discovered that would cause the results of the dynamic scenarios to be unreliably. When LiDAR sensors are moved, the point cloud becomes distorted. The distortion could, theoretically, be detected and corrected, but due to the data structure of the submitted source date of some LiDAR sensors, it was not possible to correct this distortion.

Moving a sensor at high velocities causes distortion. As the sensor traverses more distance in the time period it takes to measure a complete point-cloud, more distortion is introduced to the measured scene. This is caused by the sequential transmission of the scanned points. The time difference in between measured points allows for a relative error in the opposite direction of motion. This effect, as seen in Figure 9, is clearly visible when observing clusters of points measured on static objects lateral to the direction of movement. Early points in the point-cloud seem to be further away than points measured later in the measurement-period, resulting in the pattern representing an object being stretched along the axis of movement. With such distortion, the algorithm was not able to detect spheres in an adequate extent, even at low speeds. It was not possible to extract the distortion for some LiDARs (as discussed in Section 5 and Section 6); therefore, the decision was made to leave the results out, as they may be unrepresentative.

### 4.4. Scenario 4: Dynamic Vehicle Detection

This test was conducted with the selected top three sensors out of the previous scenarios. These three yielded best results so far and are best suited for Scenario 4:Robosense M1;Velodyne Velarray H800;Livox Horizon.

#### 4.4.1. Position Difference Test 
10
km/h

Figure 10 shows the difference between the distance to the second vehicle measured via GPS against the distance to the second vehicle measured by each LiDAR in relation to the distance. Because the LiDARs were mounted in the front of the car roof and the GPS measurements are taken at the back axle of the measurement vehicle (see Figure 4), an offset of about 2.5 m occurs, which leads to the curve seen in Figure 10.

Figure 10a shows the differences in the position measurements for the *Velodyne Velarray*. Because the sensor has a sample rate of 20 FPS, it generated most of the detection of the vehicle. The sensor is capped by its software driver to 200 m, which is the reason why the sensors measurements never exceeded this distance. The sensor managed a few detections up to 150 m, where detections occur more often. The results *Robosense M1* are shown in Figure 10b. This sensor is is also capped at 200 m by the sensors driver. Besides some outliers, the sensor shows overall precise and dense measurements up to 150 m. The *Livox Horizon* (Figure 10c) was the only sensor that was able to detect the car at a distance of over 250 m. Besides the overall good result in range, the sensor measured the position less accurately and less densely than its competitors.

#### 4.4.2. Height Measurement Test 
10
km/h

Figure 11 shows the height of the second standing vehicle measured by the LiDAR in relation to the distance. The actual height of the measured car was 1.7 m. Height measurements are less important for automotive use cases; hence the vertical resolution is often compromised. A low vertical resolution leads to a phenomenon called quantization. At far distances of around 150 m, just a few points reflect from the measured second vehicle. The result of the height measurement relies on the highest point that was reflected and is therefore only an approximation that will get better the closer the LiDAR gets to the measured object. The same applies to width measurements, even though the horizontal resolution is often higher.

The *Velodyne Velarray* has a very low vertical resolution. However, for an automotive sensor, this is nonetheless acceptable. At a distance of approximately 120 m, the points are 80 cm apart vertically, confirming the pattern seen in Figure 11a, as the probability at this distance that a point of a higher scan line does hit the standing vehicle increases. The *Robosense M1* has a wider vertical FOV alongside a higher vertical resolution than the ’Velodyne Velarray’, being less prone to the quantization problem in the vertical direction. Other than its competitors does the *Livox Horizon* not use a common grid-like scan pattern. The deployed scan pattern has benefits, as the probability is increased that some points hit the second vehicle at a larger height. Figure 11c does not show the same pattern as previously seen with the other LiDARs, which have a unique scan pattern.

#### 4.4.3. Width Measurement Test 
10
km/h

Figure 12 shows the width of the second standing vehicle measured by the LiDAR in relation to the distance. The actual width of the measured car is 1.9 m. In traffic, width and length of objects are often more important, because cars can only move in a 2D space. The quantization phenomenon explained previously will again have an impact on the measurements. It has to be noted that the LiDAR systems detect the second vehicle standing on the side, not directly from behind. Therefore the sensors mostly measure the true width, but some beams will hit the side of the vehicle as well, which falsely causes a higher width. This explains why all tested LiDAR systems measurements of the vehicle are a bit wider than they actual are.

The width measurement of the *Velodyne Velarray* is more accurate than its height measurement. Its measurements are consistent, taking the quantization pattern into account. Compared to Velarray, the *Robosense M1* is closer to the real width of the second vehicle at a distance of 80 m. This makes Robosense and Velarray close competitors in this test. Finally, the detection of *Livox Horizon* does not follow the quantization pattern and has a lot of variations. This could be explained by its scan pattern: sometimes, the beams scan just a part of the width of the car, but going up or down before scanning the whole car, causing the less accurate measurements and out-of-place-looking points below 1 m in width. Additionally, it is the only sensor that is able to return a width near ground truth at over 200 m.

## 5. Discussion

In this section, we will discuss the observations we made during the benchmark. Further, we will discuss a number of problems that occurred with the tested sensors.

Based on the results of the first scenario, static spheres increasing distances (Section 4.1), we decided to exclude three sensors from dynamic scenarios. Both the Blickfeld Cube and the Innoviz Pro were not able to perform the benchmark at a distance of 25 m and the Blickfeld cube range showed that the narrow FOV would be too narrow for the dynamic scenarios.

In Scenario 2 (Section 4.2), we identified a problem which led to poor performance in measuring the *x* direction of most sensors. This fringing problem at the edges of the plane, referred to in Section 4.2, was documented and forwarded to the manufacturers. At this point, the reason for this phenomenon is unclear and just speculative.

Scenario 3 (Section 4.3) could not be executed because of the distortion problem. This problem is inherent for all LiDARs and will affect sensors with low refresh rates more than sensors with high refresh rates. The distortion can be removed by calculating the movement of the sensor against the delta between points, but the data structures of some point clouds have dynamic sizes, rendering this approach impossible. Here, it needs to be mentioned that Livox provides an open source software solution to remove the distortion by deploying the integrated IMU of the sensor.

In the last scenario (Section 4.4), the obvious observation was made that for large distances, the fixed gaps between the scan lines of the scan patterns makes precise object detection difficult. The Livox Horizon LiDAR sticks out because of the unconventional scan pattern. With this, it outperforms its competitors in terms of range, while, on the other hand, losing precision in comparison to Robosense M1 and Velodyne Velarray H800. Scan patterns and point density also make a difference when measuring the dimension of objects. The observation we made here was that the unconventional scan pattern design of Livox can help to obtain a better approximation for the real dimensions of the scanned object.

Sensors designed for automotive applications tend to trade off vertical against horizontal FOV as can be seen with the Velodyne Velarray H800. Summarizing, we found that the information of the properties given in the data sheet is of course important and valuable, but it does not necessarily yield reliable information on how this sensor works in practice.

Another observation was that unconventional scan patterns have their benefits. LiDAR sensors have a low resolution in comparison to cameras; with conventional grid-like scan patterns, a low resolution results in big gaps between the beams. Unconventional scan patterns can counteract these gaps.

During the evaluation of the LiDAR sensors, we encountered multiple possible problems with the data, which prevent or may influence the performance of the sensors. Some of these occurred only on devices from specific manufacturers, while others can be corrected in post-processing steps.

All of the analyzed sensors had a certain level of *noise*. The noise observed on individual point measurements deviated in many cases from the expected location of where a single beam of the sensor would reflect off. In other cases, the location of the measured points was contradictory. Points were measured in mid-air with no obvious obstacle in sight. Specifically in medium and far-away regions of the point cloud, the points contained faulty or non-usable information. This can be a hazard if the algorithms used are supposed to detect patterns and objects from distant measurements. The faulty information could occasionally form clusters of points, which can be similar to a search pattern. For example, in a collision avoidance system, false positives due to noise can endanger the life of passengers if safety precautions, such as the fastening of a seat belt, are not taken. A false positive in the near field could result in an hard braking manoeuver. Not only could this cause damage to the passengers, but it could also potentially lead to loss of control over the vehicle by the driver.

The edge fringing effect of different LiDAR sensors shown in Figure 8 and explained in the respective section can be compensated through the configuration of the individual firmware. Different measurement types are available to decide by which strategy the reflected beams will be evaluated. The measurement type “strongest return” results in the reflected beams with the most intensity to be chosen for the calculation of a 3D point. Another measurement type “first return” leads to the first reflected beam to be chosen for the same calculation.

We want to point out that some problems should be addressed by the manufacturers; when asked about the fringing and distortion problem, the manufacturer’s technological support was often unaware that such problems exist or had to speculate.

## 6. Conclusions

Deploying LiDAR technology has major benefits in many robotics application fields, as well in the field of automated driving. Series production of automated vehicles would demand sensors that are precise and in an acceptable price range. A set of different LiDAR sensors were selected to be benchmarked against each other in four different scenarios. To the best of our knowledge, benchmark scenarios based on real-life use cases have not been proposed in the literature before. We divided the scenarios into static and dynamic tests. In static scenarios, both the measured object and the sensor did not move; in the dynamic scenarios, the sensor was placed on a vehicle that drove toward the measured object. In contrast to other benchmarks such as [11], the selected LiDARs were mainly based on solid-state LiDAR technology.

The findings in this paper have shown that there are considerable differences in LiDAR technologies: for individual use cases, the whole package has to be considered, including the availability and the kind of driver software. As additional software may also be required and useful, open source software should be the first choice.

Scenario 2 (*static spheres in increasing distances*, Section 4.1) and especially the *static square meter reference plane* scenario in Section 4.2 show that the tested LiDAR sensors can have major deviations in point precision.

Scenario 4 (*dynamic vehicle detection*, Section 4.4) shows that the scan pattern of a LiDAR can make a difference, an observation of which researchers and developers seem not to be very aware. When designing a LiDAR-based use case, the scan pattern should not be ignored. The results of this publication help to select the best-suited LiDAR for a particular application. Further, the minor and major differences between the tested LiDAR sensors and their particular technologies become apparent.

## Figures and Tables

**Figure 1 sensors-22-07146-f001:**
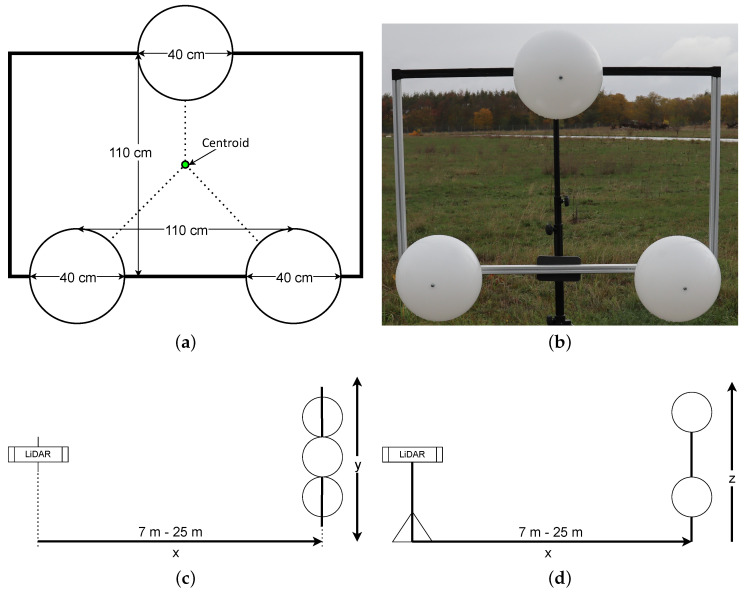
Scenario 1 setup: (**a**) Construction sketch. (**b**) Picture of the construction. (**c**,**d**) Topview and sideview of the measurement setup.

**Figure 2 sensors-22-07146-f002:**
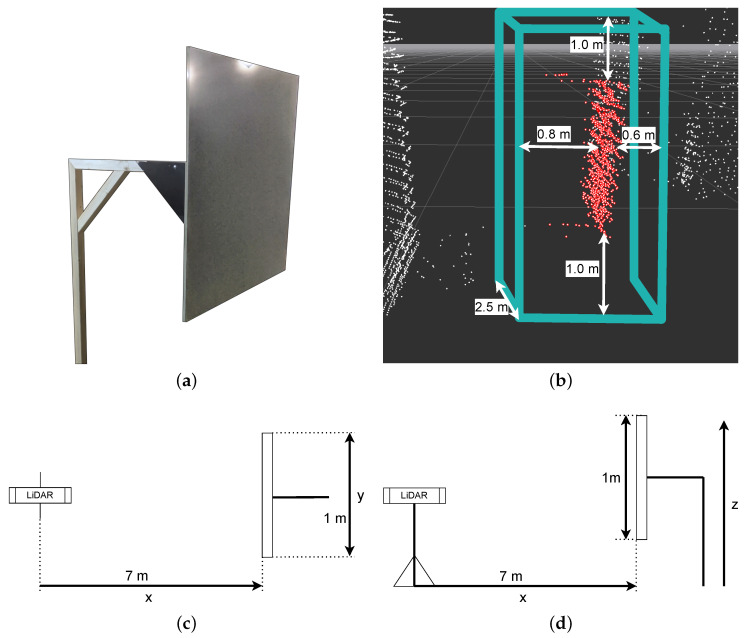
Scenario 2: (**a**) Square meter plane target. (**b**) Point cloud of a detected target. Red points are points left over after cutting out a box around the plane. The box is spaced 1 m below and 1 m above the plane, 60 cm in front and 80 cm behind. It is 2.5 m wide around the center of the plane. (**c**,**d**) Topview and sideview of the measurement setup.

**Figure 3 sensors-22-07146-f003:**
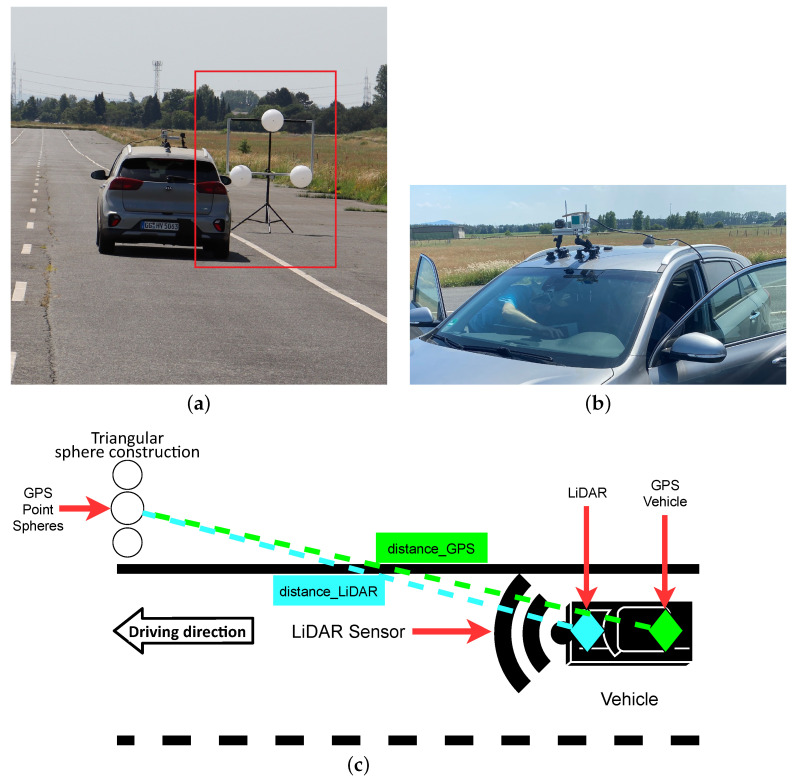
Scenario 3: (**a**) Spheres on the test track; (**b**) LiDAR mounted to the test vehicle; (**c**) Topview of the dynamic test scenario.

**Figure 4 sensors-22-07146-f004:**
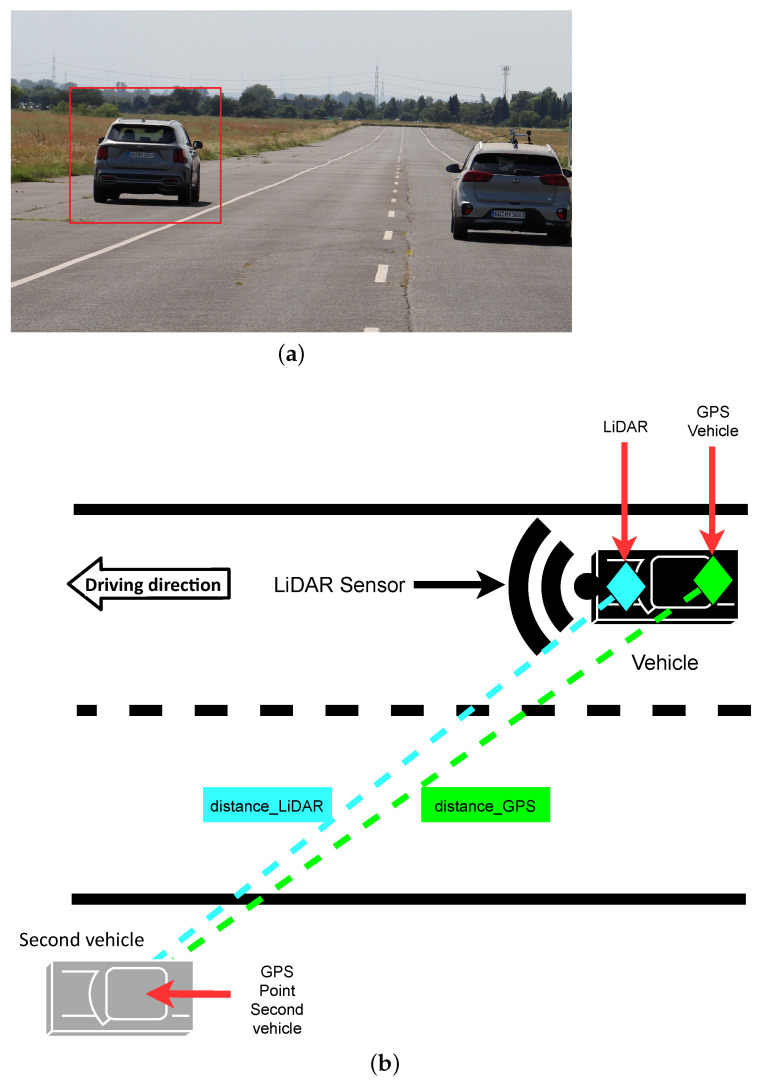
Scenario 4 (**a**) A second still-standing vehicle is the detection target. (**b**) Top view of the Scenario 4.

**Figure 5 sensors-22-07146-f005:**
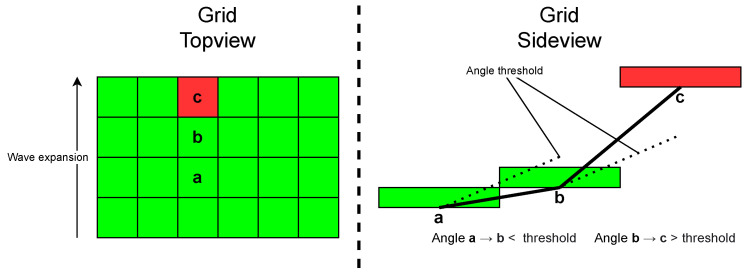
Ray-based ground–object segmentation detection functionality, sketch. The angle between cell a and b does not exceed the threshold, cell b is marked as ground cell. The angle between cell b and c exceeds the threshold, cell c is marked as non-ground.

**Figure 6 sensors-22-07146-f006:**
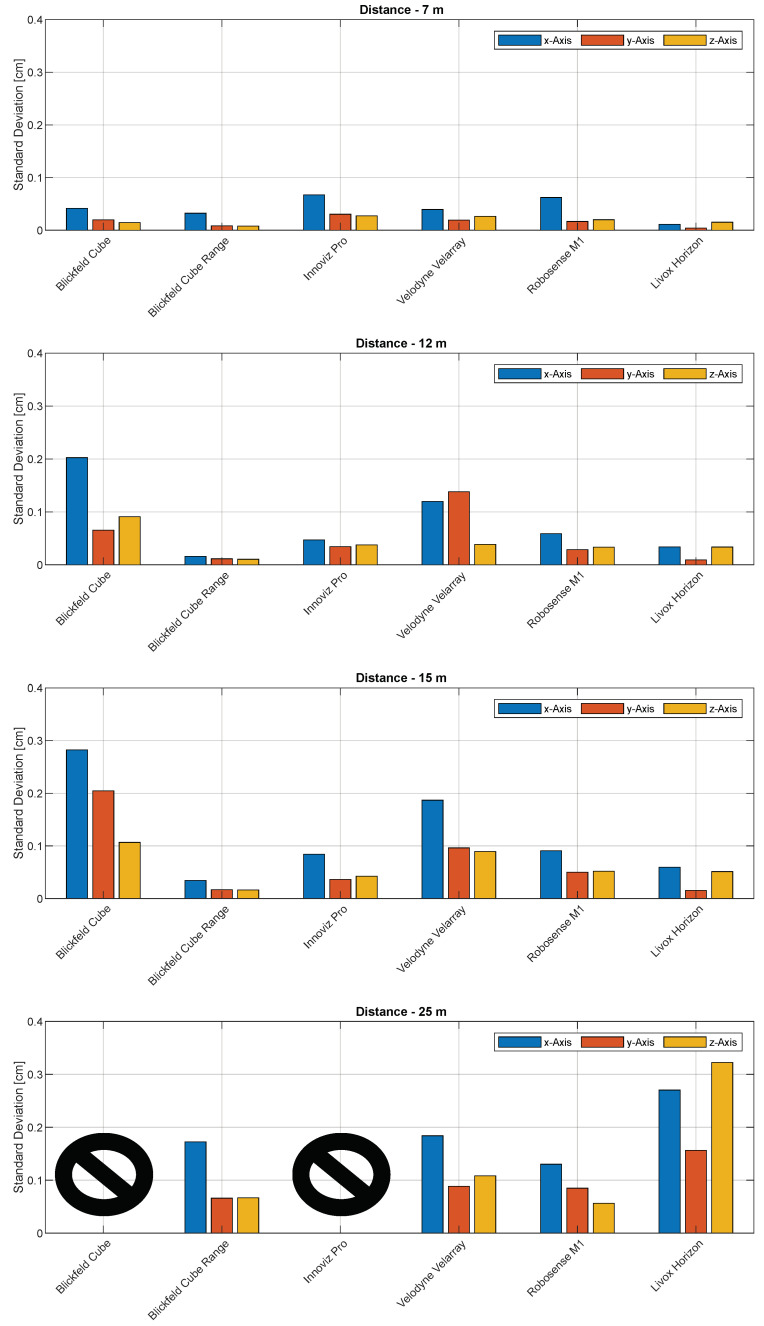
Results: Standard deviations for each individual axis in scenario ’Static Spheres Increasing Distances’ at 7 m, 12 m, 15 m and 25 m; normal distribution with 1000 samples.

**Figure 7 sensors-22-07146-f007:**
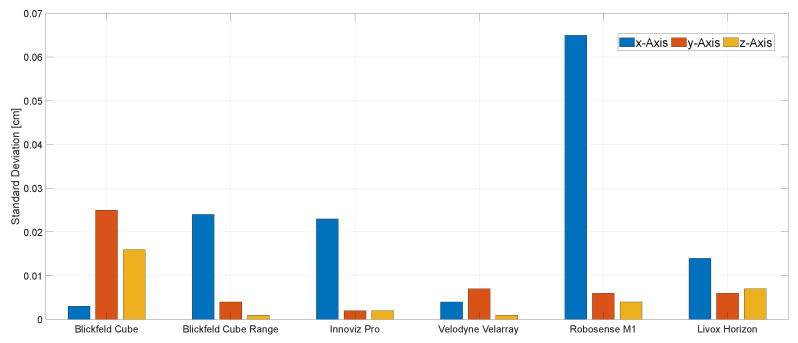
Results: Standard deviations for each individual axis in scenario ’Static Square Meter Reference Plane’ normal distribution, 1000 samples.

**Figure 8 sensors-22-07146-f008:**
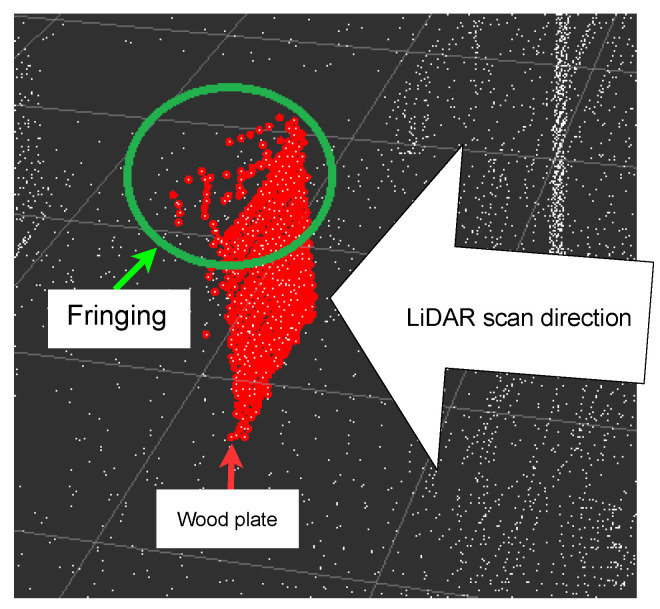
The red point cloud contains the reflections of the reference plane. The fringing at the top is noticeable (marked in the green circle), which is also known as edge fringing.

**Figure 9 sensors-22-07146-f009:**
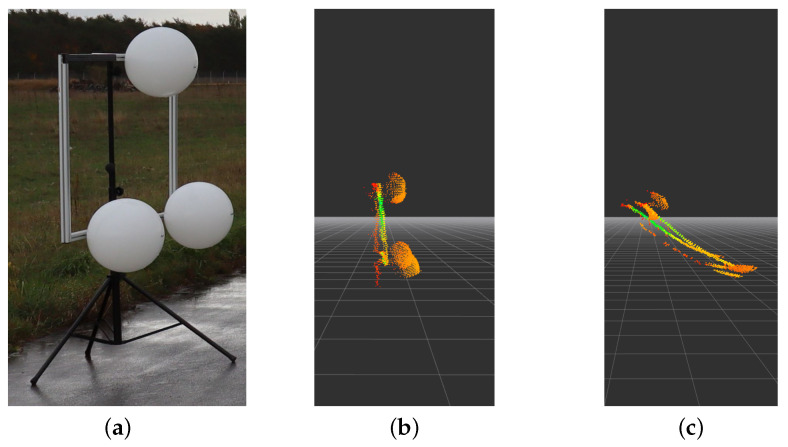
This figure presents the visualization of the distortion effect due to movement of the sensor at high velocities. (**a**) Picture of target. (**b**) View at 10 km/h, Visualization of the measured point-cloud at 10 km/h rotated so the structure can be viewed from the left side. (**c**) View at 80 km/h, The visualization of the measured point-cloud at 80 km/h rotated so the structure can be viewed from the left side. In (**c**), the measured structure is virtually stretched over a span of 2.22 m.

**Figure 10 sensors-22-07146-f010:**
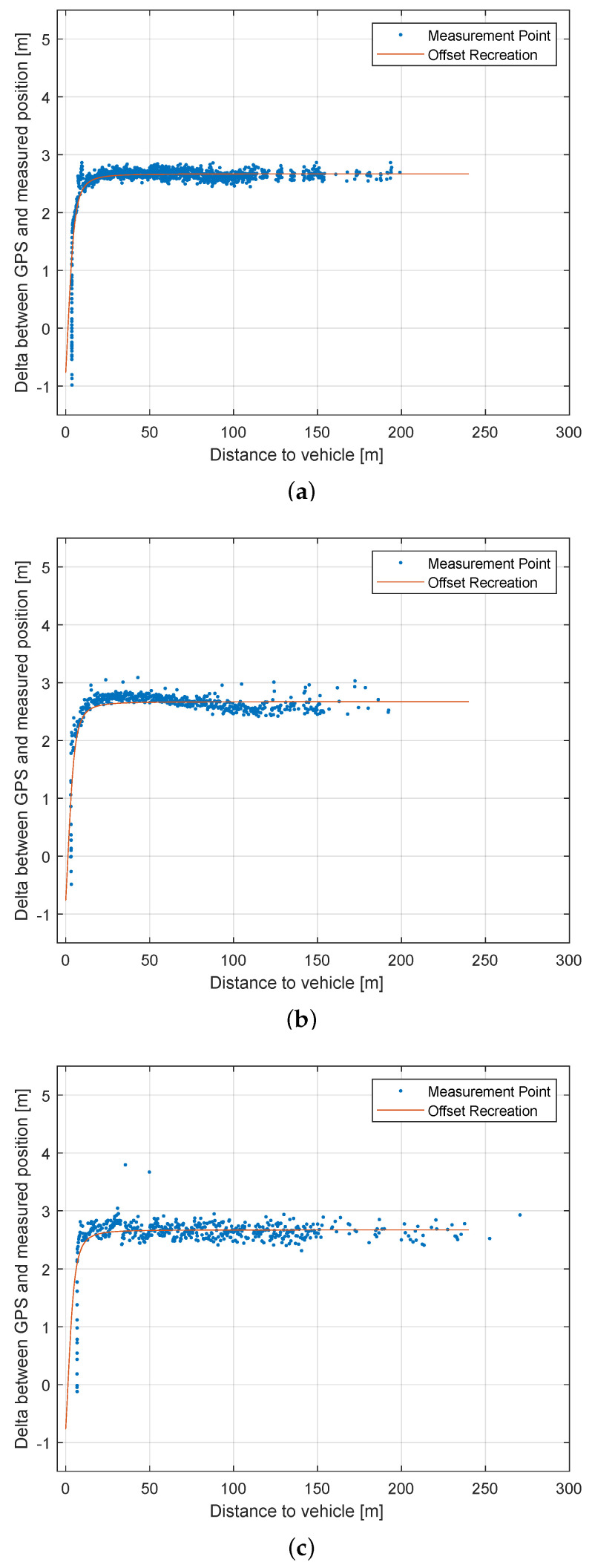
Results of Scenario 4. The difference between GPS and LiDAR measurement in relation to the distance is shown (**a**) Velodyne Velarray (1175 vehicle detections); (**b**) Robosense M1 (522 vehicle detections); (**c**) Livox Horizon (468 vehicle detections).

**Figure 11 sensors-22-07146-f011:**
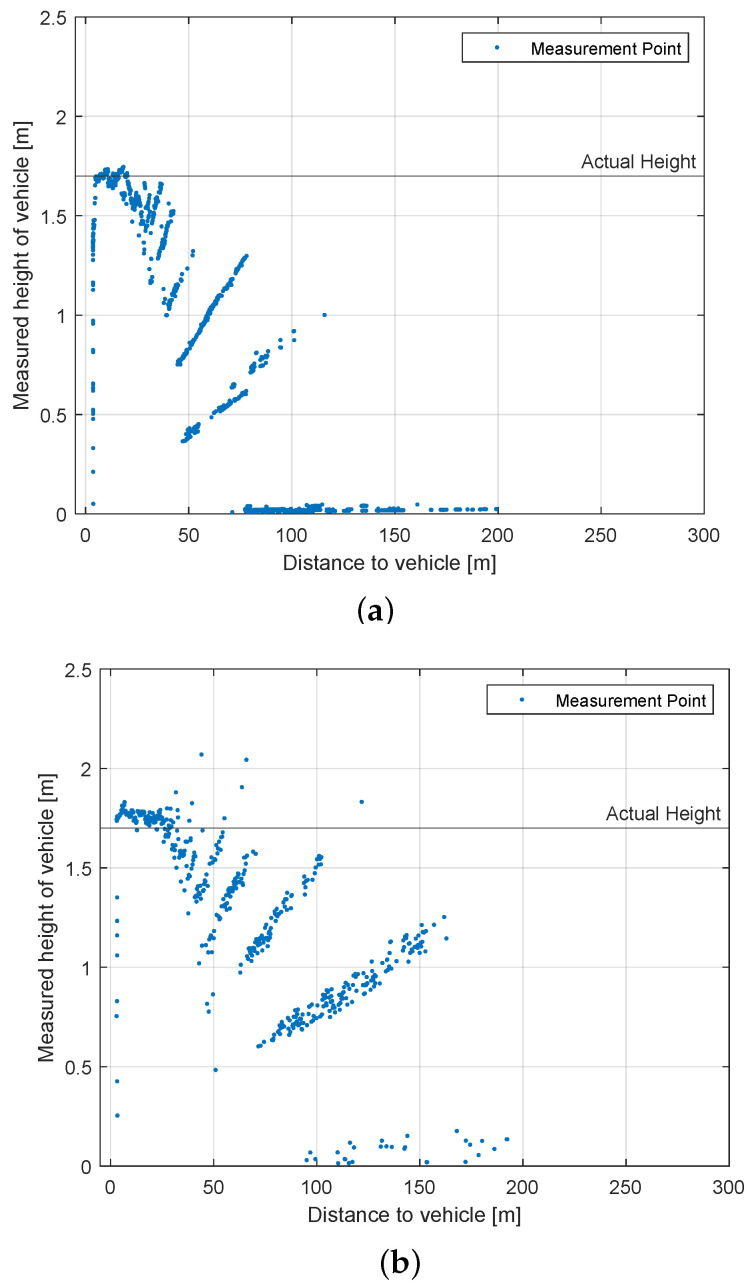
Results for Scenario 4. Measured height in relation to the distance to the second vehicle. (**a**) Velodyne Velarray (1175 vehicle detections); (**b**) Robosense M1 (522 vehicle detections); (**c**) Livox Horizon (468 vehicle detections).

**Figure 12 sensors-22-07146-f012:**
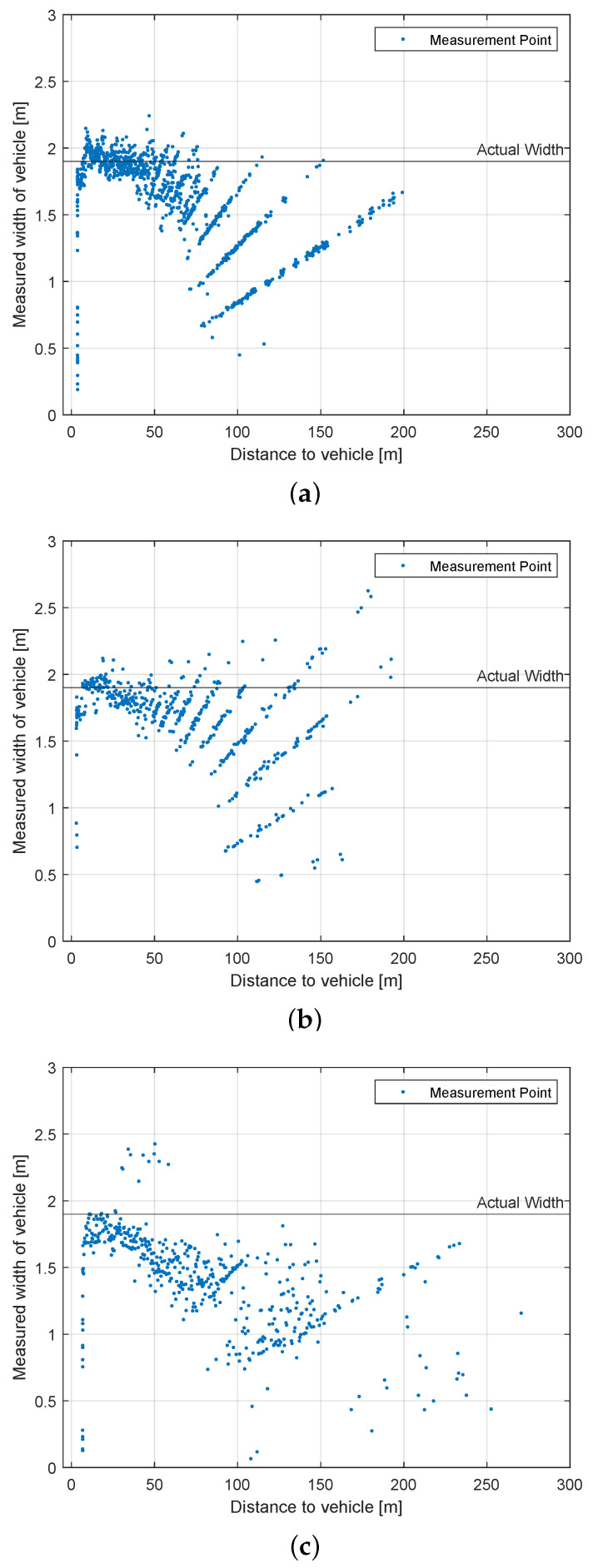
Results: Scenario ’Dynamic Vehicle Detection’. Measured width in relation to the distance to the second vehicle. (**a**) Velodyne Velarray (1175 vehicle detections); (**b**) Robosense M1 (522 vehicle detections); (**c**) Livox Horizon (468 vehicle detections).

**Table 1 sensors-22-07146-t001:** The tested sensors with their respective scanning patterns. The pictures of the scan patterns were made by facing the LiDAR toward a white wall. The screenshots were taken in the point cloud viewer from the LiDAR perspective.

	Livox	Robosense	Blickfeld	Blickfeld	Velodyne	Innoviz
	Horizon	M1	Cube	Cube Range	Velarray H800	Pro
Picture	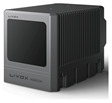	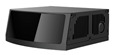	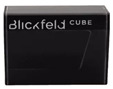	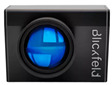	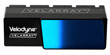	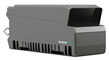
Scan pattern	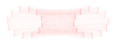	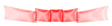	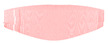	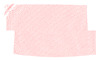	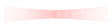	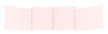
Framerate	10 Hz	10 Hz	6.3 Hz	5.7 Hz	25 Hz	16 Hz
Points per Frame	24.000	78.750	8.829	7.599	16.181	15.500
FOV	81.7° H, 25.1° V	120° H, 25° V	72° H, 30° V	18° H, 12° V	120° H, 16° V	72° H, 18.5° V
Principle	Rotating Prisms	MEMS	MEMS	MEMS	Solid State	MEMS

## Data Availability

Data available on request due to privacy restrictions. The data presented in this study are available on request from the corresponding authors. The data are not publicly available due to company policies.

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
