# Peer review of "Benchmarking of Various LiDAR Sensors for Use in Self-Driving Vehicles in Real-World Environments"

_sensors, 2022, doi:10.3390/s22197146_

Round 1

Reviewer 1 Report

In this work, authors have mentioned the benchmarking of various LiDAR sensors (Livox Horizon; 1 Robosense M1; Blickfeld Cube; Blickfeld Cube Range; Velodyne Velarray H800 and Innoviz Pro) . The overall structure of the manuscript is okay. However, I have some following comments: 

a) In Introduction section, significance of using the mentioned LiDAR sensors needs to be established. The author should mention the importance of the LiDAR sensors and mentioned the contributions in this section. 

b) The author should provide the tabular comparison of some recent works on the mentioned LiDAR sensors. 

c) The English language needs to be improve throughout the manuscript. 

d) The resolution of the figures from Fig. 14 to 16 needs to be improved. 

Author Response

In this work, authors have mentioned the benchmarking of various LiDAR sensors (Livox Horizon; 1 Robosense M1; Blickfeld Cube; Blickfeld Cube Range; Velodyne Velarray H800 and Innoviz Pro) . The overall structure of the manuscript is okay. However, I have some following comments: 

a) In Introduction section, significance of using the mentioned LiDAR sensors needs to be established. The author should mention the importance of the LiDAR sensors and mentioned the contributions in this section. 

Answer: 

We reworked introduction, and  added text to further explain the importance of the used LiDAR and the use of LiDAR in general

b) The author should provide the tabular comparison of some recent works on the mentioned LiDAR sensors. 

Answer: 

If we understood correctly, the reviewer was looking for an overview of related work where the different LiDAR sensors have been deployed. While such an overview is surely interesting, we decided against this comparison as some sensors are more popular than others and are available for a longer time than others showing an unbalanced picture of the different sensors.

c) The English language needs to be improve throughout the manuscript. 

Answer:

Throughout the whole manuscript, English was improved, and grammatical errors were corrected

d) The resolution of the figures from Fig. 14 to 16 needs to be improved.
Answer:

We enlarged all figures that were noticed as too small. Further, we ensured that all graphs and diagrams are vectorized.

Reviewer 2 Report

In this paper 6 LiDAR sensors are tested in 4 different scenarios (two static and two dynamic ones).

In Scenario 1 is not clear how the 4 distances were selected. For both dynamic scenarios (Scenario 3 & 4) is not clear the choise of vehicle's velocity.  Finally, the discussion and the conclusions sections seem to be overlapping. It seems that the major part of consclusions section should be in the discussion section.  

As far as the format is concerned, the size for Figures 4(c), 5, 7, 10, 13, 14, 15 and 16 should be increased.

Author Response

In this paper 6 LiDAR sensors are tested in 4 different scenarios (two static and two dynamic ones).

a) In Scenario 1 is not clear how the 4 distances were selected. For both dynamic scenarios (Scenario 3 & 4) is not clear the choise of vehicle's velocity.  Finally, the discussion and the conclusions sections seem to be overlapping. It seems that the major part of consclusions section should be in the discussion section.

Answer:

Explanations were added where we stated our choice selected distances for scenario one. The minimum distance was chosen because at 7 m all LiDAR had all spheres in their FOV the maximal distance was chosen after noticing that at this range the first sensors were unable to detect the spheres and knowing the performance at 25 m was a good foundation for the following dynamic scenarios. The decision to use 12 m and 15 m in between instead of equidistant distances have no particular reasons. Further, we explained our choice to make the dynamic tests at 10 km/h. The speed of was chosen 10 km/h for both dynamic scenarios due to the refresh rates of the sensors. Other tests with 80 km/h were made but were later excluded from both dynamic scenarios, because of a distortion problem discussed later in the paper. Part of the conclusion was merged into the discussion part, the overlapping part was rewritten.

b) As far as the format is concerned, the size for Figures 4(c), 5, 7, 10, 13, 14, 15 and 16 should be increased.
Answer:

We enlarged all figures that were noticed as too small. Further, we ensured that all graphs and diagrams are vectorized.